# Training Recurrent Neural Networks for BrdU Detection with Oxford Nanopore Sequencing: Guidance and Lessons Learned

**DOI:** 10.3390/genes16111356

**Published:** 2025-11-10

**Authors:** Haibo Liu, William Flavahan, Lihua Julie Zhu

**Affiliations:** 1Department of Molecular, Cell and Cancer Biology, University of Massachusetts Chan Medical School, 364 Plantation Street, Worcester, MA 01605, USA; haibo.liu@umassmed.edu (H.L.); william.flavahan@umassmed.edu (W.F.); 2Program in Molecular Medicine, Department of Genomics and Computational Biology, University of Massachusetts Chan Medical School, Worcester, MA 01605, USA

**Keywords:** machine learning, deep learning, recurrent neural network, gated recurrent unit, TensorFlow, BrdU, modified base, Oxford nanopore sequencing

## Abstract

**Background/Objectives**: BrdU (5′-bromo-2′-deoxyuridine), a synthetic thymidine (T) analog, is widely used to study cell proliferation and DNA synthesis. To precisely identify where and when DNA replication starts and terminates, it is essential to determine the BrdU incorporation rate and sites at a single-nucleotide resolution. Although several deep learning-based methods have been developed for detecting BrdU using Oxford nanopore sequencing data, there is a lack of accessible, easy-to-follow tutorials to guide researchers in preparing training data and implementing deep learning approaches as the nanopore sequencing technologies continue to evolve. **Methods**: Due to the lack of ground truth BrdU-positive data generated on the latest R10 flow cells, we prepared model training data from legacy R9 flow cells, consistent with existing tools. We processed publicly available synthetic and real nanopore DNA sequencing datasets, with and without BrdU incorporation, using a combination of open-source and custom software tools. Subsequently, we trained bidirectional gated recurrent unit (BiGRU)-based recurrent neural networks (RNNs) for BrdU detection using the TensorFlow library on the Google Colab platform. **Results**: We trained BiGRU-based RNNs for BrdU detection with a high specificity (>94%) but a moderate sensitivity due to limited BrdU-positive data. We detail the setup, training, testing, and fine-tuning of the model using both synthetic and real DNA sequencing data. **Conclusions**: Though the models were trained with data generated on legacy flow cells, we believe that this detailed protocol, covering both data preparation and model development, can be readily extended to R10 flow cells and basecallers for other base modifications. This work will facilitate the broader adoption of deep learning neural networks in biological research, particularly RNNs, which are well suited for modeling sequential and time-series data.

## 1. Introduction

DNA replication is a fundamental process essential to all living organisms. In eukaryotes, genomic DNA replication initiates at multiple bi-directional replication forks formed at stochastically activated replication origins [1]. Under normal conditions, these replication forks progress divergently, with the unwound DNA strands being copied by DNA polymerases in a semi-conservative manner until the forks meet. However, in response to various forms of replication stress, replication forks can be actively stalled, remodeled, processed, protected, and subsequently restarted [2]. DNA replication is tightly regulated to maintain genome stability, coordinate cell cycle progression, and ensure the faithful transmission of genetic information [3]. It is fundamental to cell division, growth, development, tissue repair, and the inheritance and variation in traits. Disruptions in DNA replication or associated repair pathways can lead to genomic instability, uncontrolled cell proliferation, and other cellular dysfunctions, thereby contributing to the onset and progression of various diseases. Consequently, a thorough understanding of the mechanisms and regulation of DNA replication is crucial for identifying therapeutic targets and advancing the development of more effective, personalized treatment strategies—particularly in cancer [4,5].

BrdU (5′-bromo-2′-deoxyuridine) is a synthetic thymidine analog with low cytotoxicity, which can be incorporated into newly synthesized DNA during replication or repair by DNA polymerase, substituting for thymidine [6,7,8]. Consequently, BrdU has been widely used both in vitro and in vivo to study cell cycle kinetics, DNA replication, DNA repair, sister chromatid exchange, and to assess proliferation of healthy and pathological cells or tissues under various conditions [8,9,10,11]. However, until recent years, the detection of BrdU incorporated into DNA has been limited to low-throughput and/or low-resolution methods, such as DNA fibering assays with anti-BrdU antibodies [11] and BrdU immunoprecipitation followed by either microarray hybridization (BrdU-IP-chip or E/L Repli-chip) [12,13] or high-throughput Next Generation Sequencing (qBrdU-seq or E/L Repli-seq) [14,15]. These methods provide limited temporal and spatial resolution due to the challenges in precisely identifying the genomic positions of BrdU incorporation. To enable the detailed analysis of the DNA replication process, it is desirable to accurately detect and quantify BrdU along extended stretches of single DNA molecules at a single-nucleotide resolution [7,16,17,18].

Advances in third-generation sequencing technologies, such as the PacBio Single Molecule Real Time (SMRT) sequencing and the Oxford Nanopore Technologies (ONT), have enabled the detection of BrdU and a variety of other modified bases and base analogs in long single DNA molecules at higher resolutions [16,19,20,21,22]. PacBio SMRT detects these modifications by leveraging the characteristic changes in DNA polymerase kinetics during the incorporation of modified nucleotides [23,24]. In contrast, ONT identifies base modifications by analyzing ionic current signals generated as single-stranded DNA fragments pass through individual nanopores. The ionic current signals are influenced not only by the nucleotide passing through the nanopore but also by several adjacent nucleotides. Together, these bases form a short, fixed-length sequence known as a nucleotide *k*-mer. The size of *k*-mer varies with nanopore flow cell versions—5 for R9 and 9 for R10. Many modified bases or base analogs cause detectable deviations in current signals as they pass through the nanopores, compared to their unmodified or natural base counterparts [21,22]. By leveraging these characteristic signal patterns, several deep learning models have been developed to identify both modified and unmodified bases (for reviews, see [21,22]). RepNano is a convolutional neural network (CNN)-based basecaller specifically designed to detect BrdU incorporation [17]. Its input is a five-dimensional vector: the first four dimensions represent the base identity using a one-hot encoding derived from a standard basecaller such as Dorado (https://github.com/nanoporetech/dorado, (accessed on 1 November 2025)), while the fifth dimension encodes the normalized current shift across each 96-nucleotide segment. RepNano applies a 96-nucleotide CNN window that slides by 10 nucleotides across the read; for each 10-nucleotide segment, it averages the CNN’s BrdU predictions from all the overlapping 96-nucleotide windows covering that segment. Due to its design, RepNano detects BrdU incorporation at a 10-nucleotide resolution, rather than at a single-nucleotide resolution. DNAscent v2 and its successors [16,25,26], are BrdU basecallers built on residual convolutional neural networks. They use signal-level alignments generated by a Hidden Markov Model (HMM) to predict BrdU incorporation with a single-nucleotide resolution [16]. NanoForkSpeed [18] is a pipeline for inferring the velocity of individual DNA replication forks based on BrdU incorporation sites and content. It leverages the ONT’s Megalodon (https://github.com/nanoporetech/megalodon (accessed on 1 November 2025), obsolete) supplemented with a flip-flop recurrent neural network (RNN) model, which was trained with the Taiyaki (https://github.com/nanoporetech/taiyaki (accessed on 1 November 2025), obsolete) to perform BrdU basecalling. Notably, all existing models were trained using data generated on R9 flow cells due to the unavailability of R10 data, although DNAscent (v4.1.1) claims its compatibility with data produced by Oxford R10.4.1 flow cells.

Because the ONT’s current signal at each nucleotide is influenced not only by the nucleotide itself but also by its flanking nucleotides (i.e., *k*-mer), a couple of state-of-the-art basecallers, both general-purpose and modification-specific, have been developed to exploit this context dependency. These models commonly incorporate bi-directional RNNs with long short-term memory (LSTM) units or gated recurrent units (GRUs) for improved performance [27,28,29,30,31]. Although residual CNNs, as implemented in DNAscent2, inherently capture some degree of context dependency, no existing BrdU detection tools fully exploit more advanced sequence modeling approaches such as bidirectional GRU (BiGRU)-based RNNs. We therefore hypothesized that a BrdU basecaller leveraging a BiGRU-based RNN could enhance detection accuracy by more effectively capturing the complex, context-dependent patterns in nanopore signals. Here, we present DeepBrdU, a BrdU basecaller, built on a three-layer, BiGRU-based RNN with a timestep of five. The input to a GRU is a 7-dimensional vector: the first three dimensions represent the normalized mean, standard deviation, and dwell time of the current signal corresponding to the associated 5-mers; the remaining four encode the identity of the base at the middle position by one-hot encoding for A, C, G, and T. Due to the unavailability of ground truth BrdU-positive data generated on the latest R10 flow cells, like all existing tools, we prepared model training data from legacy R9 flow cells. Although the final model does not outperform the existing BrdU basecallers, primarily due to limited training data, we provide a detailed guide for data preparation and training of multi-layer, BiGRU-based RNNs. This framework can be adapted to develop basecallers for other modified bases or base analogs.

## 2. Materials and Methods

### 2.1. Nanopore Sequencing Data from Synthetic 5-mer Reference Standards

The sequencing data (Dataset I) was generated by the Flavahan’s lab (a coauthor of this study) and is publicly available in the Sequence Read Archive (SRA; Accession number SRX23943693). Briefly, synthetic oligodeoxyribonucleotides, each five nucleotides long, were designed to represent all possible combinations (5^5^ = 3125) of five bases: A, BrdU, T, C, and G. These synthetic sequences, referred to as “5-mers”, were produced using a barcoded split-pool synthesis (BSPS) strategy and sequenced on a MinION Mk1C instrument on an R9.4.1 flow cell for 24 h [32]. The sequence of each 5-mer was determined by the order and identity of five consecutive barcodes incorporated into the sequencing template. The detection frequencies of these 5-mers by nanopore sequencing are shown in Appendix A.

### 2.2. Nanopore Sequencing Data from Plasmid DNA Synthesized by Primer Extension

The sequencing data (Dataset II) were downloaded from the SRA (Accession numbers ERS4418251 and ERS4418252). Briefly, primer extension reactions were performed using linearized and denatured pTYB21 plasmid DNA (New England Biolabs, Catalog # N6709S) as templates and dATP, dGTP, dCTP, and BrdUTP (BrdU-labeled sample; ERS4418252) or dTTP (BrdU-free control; ERS4418251) as substrates [17]. The remaining single-stranded DNA was enzymatically removed. The resulting double-stranded DNA—containing one strand synthesized by primer extension—was purified and prepared for sequencing using a 2D low-input kit with the R9 chemistry. Raw reads were basecalled using Metrichor (v2.40.17) by Hennion et al. [17]. As basecalled data were already included in the downloaded FAST5 files, no additional basecalling was performed for this dataset.

### 2.3. Nanopore Sequencing Data from Mouse and Human Cell Lines

The sequencing data (Dataset III) were generated by the Flavahan’s lab (a coauthor of this study) and is publicly available in the SRA (Accession number SRX28421100). Briefly, DNA was extracted from human 293T cells cultured in medium containing 100 µM of BrdU, resulting in approximately 20% of substitution of thymidine with BrdU (i.e., a 20% BrdU substitution rate), and from mouse 3T3 cells cultured in BrdU-free medium. Nanopore sequencing libraries were prepared for each sample using the Rapid Barcoding Kit (mouse 3T3—barcode 1, human 293T—barcode 2), pooled, and co-sequenced on a MinION MK1C with R9.4.1 flow cells for 24 h. For model training and testing, one multi-read FAST5 file was selected for each cell line: FAS54789_pass_barcode01_dc523426_25.fast5 for the mouse 3T3 cells and FAS54789_pass_barcode02_dc523426_14.fast5 for the human 293T cells. Each FAST5 file contained 4000 reads.

### 2.4. Nanopore Sequencing Data from Yeast Cultures

The sequencing data (Dataset IV) was obtained from the SRA (Accession number ERS10113943, part of BioProject PRJEB50302) and was originally generated by Theulot et al. [18]. Briefly, thymidine-auxotroph MCM869 yeast strain was cultured overnight in synthetic complete medium containing 100 μM thymidine, then washed twice to remove residual thymidine. Diluted yeast cells were subcultured in fresh synthetic complete medium supplemented with varying ratios of BrdU to thymidine (0:100; 10:90; 20:80; 30:70; 40:60; 50:50; 60:40; 70:30; 80:20; 90:10, and 100:0) and grown for 24 h. Genomic DNA was extracted for each culture condition. Indexed nanopore sequencing libraries were prepared for each sample, pooled, and sequenced using R9.4.1 flow cells. For this study, only two subsets of data—designated as BO_testing_1 and BY_testing_1—were used, corresponding to BrdU substitution rates of 0% and 80.3% [18].

### 2.5. Preprocessing Nanopore DNA Sequencing Data

A general flowchart illustrating the processing pipeline of nanopore DNA sequencing data is shown in Figure 1A. Representative scripts for preprocessing nanopore DNA sequencing data are provided in Appendix A and are also accessible on GitHub (https://github.com/haibol2016/DeepBrdU (accessed on 1 November 2025), v0.0.1).

***Basecalling and filtering***: For datasets in single-read FAST5 files, the files were first converted into multi-read FAST5 format using the single_to_multi_fast5 command of ont-fast5-api (v4.1.3). The resulting multi-read FAST5 files were then converted to POD5 format using the pod5 convert FAST5 command in POD5 (v0.3.15). For datasets already in multi-read FAST5 format, files were directly converted to POD5 format as described above. Basecalling was then performed using the POD5 files as input with Dorado (v0.9.1) and the basecalling model DNA_r9.4.1_e8_sup@v3.6. Basecalled BAM files generated by Dorado were converted to FASTQ files, using the samtools fastq command in SAMtools (v1.16.1) [33]. Quality control of FASTQ reads was performed using ToulligQC (v2.7.1). The resulting ToulligQC reports are provided in Appendix A. The resulting reads were mapped to the appropriate reference sequences using Minimap2 (v2.26) [34]: the theoretical 5-mer synthesis sequences for Dataset I (Appendix A), the reformatted pTYB21 plasmid sequence (https://www.neb.com/en-us/tools-and-resources/interactive-tools/dna-sequences-and-maps-tool (accessed on 1 November 2025)) for Dataset II, the Ensembl primary assembly of GRCm39 (mouse, GCA_000001635.9) and the Ensembl T2T-CHM13v2.0 (human, GCA_009914755.4) for Dataset III, and the Ensembl yeast reference genome R64-1-1 (GCA_000146045.2) for Dataset IV. The alignments were filtered using the samtools view command in SAMtools (v1.16.1) to retain only reads with a minimal mapping quality score of 20 (Dataset I) or 60 (Datasets II, III, and IV). Additionally, reads with mapping lengths shorter than 270 (Dataset I) or 500 bases (Datasets II, III, and IV) were removed. The remaining aligned reads were extracted from the FASTQ files using the seqtk subseq command in Seqtk (v1.4). For Dataset I, only reads containing exactly five barcodes and a valid 5-mer sequence were kept for downstream analysis.

***Resquiggling***: For Datasets I, III, and IV, which were originally stored in multi-read FAST5 format, the files were first converted to single-read FAST5 files using the multi_to_single_fast5 command from ont-fast5-api (v4.1.3). The read information from the corresponding FASTQ files was then written into each single-read FAST5 file using the tombo preprocess annotate_raw_with_fastqs subcommand of Tombo (v1.5.1). In cases where the resulting FAST5 files were incompatible with tombo resquiggle, the files were repacked using h5repack tool of HDF5 (v1.14.3). Resquiggling was performed using the tombo resquiggle subcommand of Tombo (v1.5.1) with the following parameter settings: –-DNA–processes 12 –-ignore-read-locks–overwrite–threads-per-process 4 —sequence-length-range 200 50,000 —signal-length-range 500 500,000—include-event-stdev–basecall-group Basecall_1D_000 —num-most-common-errors 5. During resquiggling, Tombo maps each read to the corresponding reference sequences (see above), normalizes current signals within reads, detects events (i.e., segments of current signals corresponding to the middle bases of 5-mers), assigns sequence to signal, and resolves skipped bases. Single-read FAST5 files updated by Tombo were retained only if events were successfully detected using custom R script. Dataset II, generated using the ONT 2D sequencing strategy, was already in the single-read FAST5 format and contained sequences for both DNA strands in the “BaseCalled_template” and “BaseCalled_complement” groups. Resquiggling for Dataset II followed a modified approach. First, the plasmid pTYB21 reference sequence was adjusted so that the primer sequence appeared at the 5′ extremity and was then trimmed. For each read, the two basecalled sequences (from the template and complement groups) were aligned to the trimmed reference using Minimap2 (v2.26). The group corresponding to the alignment with SAM flag of 0 was selected to set the–basecall-group parameter in tombo resquiggle, while the other parameters remained as previously described. The resulting FAST5 files were then filtered using the same criteria outlined above. For quality control of Dataset II, FASTQ reads corresponding to alignments with a SAM flag of 0 were extracted, and their quality was assessed using FastQC (v0.11.9). The resulting FastQC reports are provided in Appendix A.

***Feature and label engineering***: For each called base of 5-mer, three signal-derived features were computed by tombo resquiggle from the events group of the resquiggled single-read FAST5 file: normalized mean values (*f_m_*), standard deviation (*f_d_*), and event length (*f_l_*), defined as the number of current signals. In addition, the nucleotide identity assigned to each event by basecalling (A, T, C, or G) was encoded as a one-hot vector [*f_A_*, *f_C_*, *f_G_*, *f_T_*] where the element corresponding to the base was set to 1 and all others to zero. Each event was therefore represented by a seven-dimensional feature vector: *x_i_* = [*f_m_*, *f_d_*, *f_l_*, *f_A_*, *f_C_*, *f_G_*, *f_T_*], for *i* = 1, …, 5. This representation integrates both signal characteristics and base identity, providing a compact input feature set for downstream model training.

For Dataset I, the ground truth base at the middle position of each 5-mer (either T or BrdU) was determined based on the alignment of reads to the reference sequence and the known barcode combinations. For Datasets II and III, only features corresponding to 5-mers with a basecalled T at the middle position were retained for model training, validation and testing. In the mouse 3T3 cell line (Dataset III), which was cultured in BrdU-free medium, the true base at the middle position of each 5-mer was assumed to be T. In contrast, for the human 293T cell line (Dataset III), which was partially BrdU-labeled, ground truth labels were not directly available. Each ground truth label was one-hot encoded into a 5-dimensional vector: *y_i_* = [*f_A_*, *f_BrdU_*, *f_C_*, *f_G_*, *f_T_*], *i* = 1, …, 5, where a value of 1 indicates the true base identity at the middle position of the 5-mer, and 0 otherwise. This vector was used as the supervised label for the corresponding input feature vector *x_i_*. Label assignment for Dataset II (BrdU labeling by primer extension) was more complex due to the presence of double-stranded DNA molecules formed by random renaturation rather than primer extension [17]. To address this, we first trained a BiGRU-based RNN model using only Dataset I and the BrdU-free mouse data. This model was then used to estimate the BrdU substitution rate in individual reads aligned to the strand synthesized by primer extension (see **Resquiggling**). The read-level BrdU substitution rates displayed a bi-modal distribution (Figure 2E), indicating a mixture of labeled and unlabeled molecules. The unlabeled molecules most likely resulted from renatured double-stranded plasmid DNA. Reads with estimated substitution rates greater than 12.5% were considered fully BrdU-labeled. The true base at the middle position of each 5-mer from BrdU-labeled reads was assumed to be BrdU.

### 2.6. Model Training and Testing

Engineered feature-label pairs for 5-mers were prepared from three sources: Dataset I (used for Models I, II, and III), a randomly sampled 0.01% of BrdU-free mouse data from Dataset III comprising approximately 800,000 5-mers (used for Models II, and III), and BrdU-labeled data from Dataset II (used for Model III). For each model, the data were shuffled and split into training (80%), validation (10%), and test (10%) sets used to train, validate, and test for model development and evaluation. BiGRU-based RNN models were implemented using the TensorFlow framework (v2.18.0) and trained on the Google Colab platform (https://colab.research.google.com/ (accessed on 1 November 2025)) equipped with NVIDIA A100 GPUs. Each model consisted of three stacked layers of BiGRUs followed by a fully connected dense output layer (Figure 1B). Hyperparameter tuning was performed based on model’s performance on the validation set. In the best-performing model, each GRU cell contained 128 units, and the batch size of 128 was used. Training was carried out for up to 30 epochs with an early stopping callback strategy to prevent overfitting. The final trained model was evaluated on the test set, the remaining 99.99% of mouse reads from Dataset III, and the BrdU-labeled pTYB21 strand of the pTYB21 plasmid (Dataset II, when not used in training). The trained model was also applied to predict BrdU incorporation sites in human 293T cell reads (Dataset III) and yeast reads (Dataset IV).

### 2.7. Performance Metrics

Performance metrics of the models were calculated using the following equations:Accuracy = (TP + TN)/(TP + TN + FP + FN),Precision = TP/(TP + FP),Sensitivity = recall = TP/(TP + FN),Specificity = TN/(TN + FP),
where TP, TN, FP, and FN represent true positive, true negative, false positive, false negative, respectively.

All performance metrics were evaluated at a single-nucleotide resolution, with feature vectors corresponding to individual 5-mers as input and the predicted base types at the central positions of the 5-mers as output. The BrdU substitution rate was calculated at a read level as the percentage of thymidine in a read predicted as BrdU by a model.

## 3. Results

David et al. recently developed a BSPS method for generating reference standard oligonucleotides for detecting novel DNA base modifications via nanopore sequencing [32]. As part of this collaborative effort, our team led the development and training of a three-layer BiGRU-based RNN to identify BrdU incorporation. We used the full set of 3125 (5^5^) synthetic 5-mers composed of A, T, BrdU, C, and G (Dataset I), sequenced using the R9.4.1 chemistry, and supplemented the training with the BrdU-free mouse data (Dataset III). The model achieved satisfactory specificity and moderate sensitivity [32]. Here, we detail our data preparation, model development, and ongoing efforts to improve the basecalling performance.

### 3.1. Training on Synthetic 5-mers Alone Limits RNN Model Generalization

We initially hypothesized that the synthetic 5-mer dataset (Dataset I) would be sufficiently representative for training a BiGRU-based RNN to detect BrdU. A three-layer BiGRU-based RNN (Model I) trained solely on this dataset achieved strong performance on the held-out synthetic test data (accuracy: 0.9441; precision: 0.9448; recall: 0.9443; and area under precision-recall curve (PRC): 0.9867; confusion matrix shown in Figure 2A). However, when applied to BrdU-free mouse data (Dataset III), the model exhibited poor specificity, incorrectly labeling over 23% of T as BrdU (Figure 2B).

### 3.2. Augmenting with the BrdU-Free Data Improves RNN Model Specificity

To improve model performance, we supplemented the synthetic 5-mer dataset (Dataset I) with BrdU-free mouse data for training a new BiGRU-based RNN (Model II) using the same architecture. Evaluation on the held-out portion of this combined dataset showed improved performance (accuracy: 0.9588; PRC: 0.9911; precision: 0.9594; recall: 0.9585; confusion matrix shown in Figure 2C). When applied to the remaining 99.99% of BrdU-free mouse data (Dataset III), the model exhibited significantly higher specificity, miscalling only 3% of T as BrdU. However, the model’s sensitivity was limited. In human 293T data (Dataset III), where a ~20% BrdU substitution rate had been independently determined by mass spectrometry [32], the model detected a BrdU substitution rate much lower than 20% (Figure 2D). Similarly, when tested on the pTYB21 primer extension product, expected to contain 100% BrdU substitution [17], the model detected only 12.5% on average (Figure 2E), confirming its low sensitivity for BrdU bases.

### 3.3. Augmenting with BrdU-Positive Data Improves RNN Model Sensitivity

To improve the sensitivity of the RNN model for BrdU basecalling, we augmented the combined training data, composed of the synthetic 5-mer data (Dataset I) and the BrdU-free mouse data (Dataset III), with fully BrdU-substituted primer extension data (Dataset II). This BrdU-positive data contained 1018 of the 1024 (4^5^ = 1024) possible 5-mer combinations containing A, C, G and BrdU, but lacked T-containing 5-mers. The resulting model (Model III) maintained similar specificity on held-out test data (accuracy: 0.9427; PRC: 0.9852; precision: 0.9428; recall: 0.9426; confusion matrix shown in Figure 2F) and the remaining mouse data. Notably, it showed improved sensitivity on the human data with a known BrdU substitution rate of 20% (Dataset III) (Figure 2G). However, testing on a yeast dataset (Dataset IV) with a known BrdU substitution rate of 80.3% further confirmed the sensitivity of Model III is still not high enough (Figure 2H), suggesting that additional, more representative BrdU-positive training data may be required to further enhance sensitivity.

## 4. Discussion

BrdU is widely used to trace spatiotemporal patterns of DNA replication, a process important for both basic and biomedical research. Third-generation sequencing technologies, such as the ONT, offer the potential to detect BrdU as well as other base modifications at a single-base resolution. However, realizing this potential usually requires a basecaller specifically designed to detect BrdU or other modified bases and trained with high-quality data with ground-truth labels.

The ionic current signal for a nucleotide passing through a nanopore is influenced not only by the nucleotide itself but also by its neighboring nucleotides in a *k*-mer context. To capture this, we trained several three-layer, BiGRU-based RNN models for BrdU basecalling using progressively more representative datasets. Model performance improved in terms of both specificity and sensitivity as we incorporated increasingly diverse training data. Nonetheless, the sensitivity of our best model was suboptimal. This limitation is likely due to the constrained sequence context in the synthetic 5-mer dataset (Dataset I), where all possible 5-mers were flanked by the same fixed sequences. In contrast, in the real genome sequencing data generated by the ONT, 5-mers are embedded in a much broader range of sequence contexts.

Including BrdU-positive data from the fully BrdU-substituted pTYB21 strands (Dataset II), which contained 1018 unique 5-mers composed of A, C, G, and BrdU, increased the diversity of the training data and significantly improved model sensitivity. Although this dataset did not cover all 3125 possible 5-mers and was sequenced using the R9 flow cells, its diversified contexts of 5-mers contributed to enhanced generalizability of the model. We anticipate that with more representative BrdU-positive data, a highly sensitive and specific GRU-based RNN basecaller for BrdU can be developed.

We acknowledge that our DeepBrdU models for BrdU detection were trained using nanopore sequencing data generated on R9 flow cells with legacy chemistries, which have since been discontinued and replaced by R10 flow cells featuring updated chemistry. Given that the R10 flow cell incorporates an improved nanopore design with two “readers” and produces substantially different current signal patterns, our current model cannot be directly applied to data generated on R10 flow cells. We would have trained our model using BrdU-positive and negative data generated on R10 flow cells if such datasets had been available. Nevertheless, we believe that our data preparation workflow and model architecture can be readily adapted to develop a new BrdU basecaller using R10-derived training data, with only minor modifications: (1) using f5c (v1.2+) [35] or Uncalled4 [36] instead of Tombo for resquiggling, and (2) using 9-mer rather than 5-mer features for model training [37].

Based on the insights gained from this study, we propose two complementary approaches to generate representative training data on R10 flow cells for training RNN models for BrdU detection. One approach involves synthesizing 17-mer reference oligonucleotides using the BSPS method [32] such that each 9-mer is flanked on both sides by four randomly variable nucleotides, thereby better mimicking realistic sequencing contexts. A potential limitation of this approach is the reduced synthesis efficiency and increased cost with larger *k*-mer size. An alternative approach is as follows. Using DNA-free mRNA extracted from an organism with a well-sequenced, assembled and annotated genome as templates, double-stranded DNA molecules can be synthesized in which thymidine is fully replaced by BrdU. This can be achieved by performing reverse transcription and second-strand cDNA synthesis in reactions containing BrdUTP but no dTTP. The resulting cDNA library can be further normalized and amplified as described previously [38], substituting BrdUTP for dTTP in the amplification reactions. Although this second method can generate at most 4^9^ of 5^9^ 9-mers, it provides highly representative sequence contexts for those 9-mers. DNA produced by either approach can undergo standard library construction using the appropriate chemistry and be sequenced on an ONT platform with the corresponding flow cells. Together, these complementary strategies would enable the generation of more comprehensive BrdU-positive training datasets, thereby improving the sensitivity and specificity of BrdU basecallers.

## 5. Conclusions

In summary, we developed a multi-layer, BiGRU-based RNN model for basecalling BrdU in DNA sequenced using the ONT. This study provides details of the data preparation and model training processes. The model achieved high specificity but exhibited limited sensitivity, indicating opportunities for further improvement. We also proposed key strategies for generating more representative training datasets to enhance model performance. Although the models were trained using data generated on legacy R9 flow cells, we believe that this comprehensive protocol—encompassing both data preparation and model development—can be readily extended to R10 flow cells and other basecallers. This work will facilitate broader adoption of deep learning approaches in biological research, particularly RNN-based methods that are well suited for modeling sequential and time-series data.

## Figures and Tables

**Figure 1 genes-16-01356-f001:**
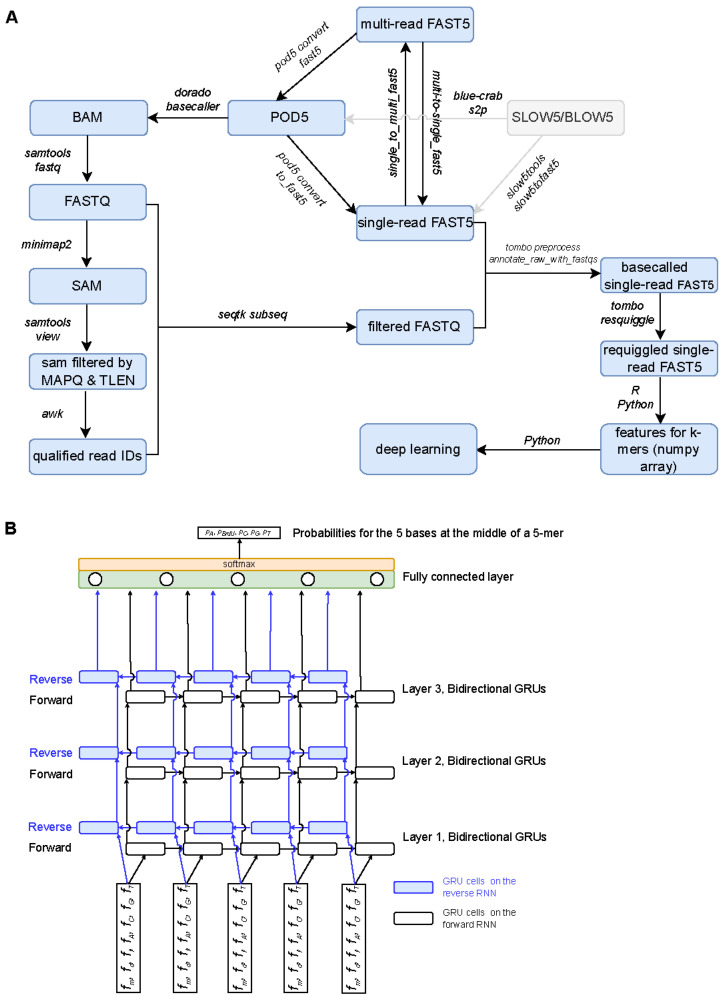
Workflow for data preparation and architecture of BiGRU-based RNNs for BrdU detection. (**A**) Flowchart depicting the data preparation pipeline. Nanopore raw sequencing data in single-read FAST5, multi-read FAST5, or SLOW5/BLOW55 formats was converted to POD5 format for basecalling, unless already provided in the POD5 format. Primary basecalling was performed using Dorado to generate BAM files, which were converted to the FASTQ format containing basecalled reads. Reads were aligned to reference sequences, and the resulting SAM files were filtered to retain only those with mapping quality (MAPQ) and alignment length (TLEN) values above defined thresholds. The identifiers of qualified reads were extracted from the filtered alignment files and used to subset the FASTQ files, producing datasets containing only high-quality reads. For data provided in multi-read FAST5, POD5, or SLOW5/BLOW55 formats, files were converted to single-read FAST5. Corresponding basecalled sequences from the FASTQ files were written into the single-read FAST5 files. Resquiggling was then performed to align ionic current signals with nucleotide positions (events) and to compute per-event statistics, including normalized mean, standard deviation, and event length. These statistics, together with base identity, were extracted from the resquiggled single-read FAST5 files and encoded as input features for deep learning. The tools and/or commands used at each stage are indicated along the arrows. Additional details are provided in the Section 2 and Appendix A. (**B**) Schematic representation of the BiGRU-based RNN architecture for BrdU basecalling. Seven-dimensional feature vectors corresponding to each base within a 5-mer were provided as input to a three-layer BiGRU-based recurrent neural network (RNN). The network outputs were passed to a fully connected layer comprising five perceptrons with a softmax activation function, producing base probabilities for adenine (A), bromodeoxyuridine (BrdU), cytosine (C), guanine (G), and thymine (T) at the central position of the 5-mer. Arrows show the data flow in the neural network.

**Figure 2 genes-16-01356-f002:**
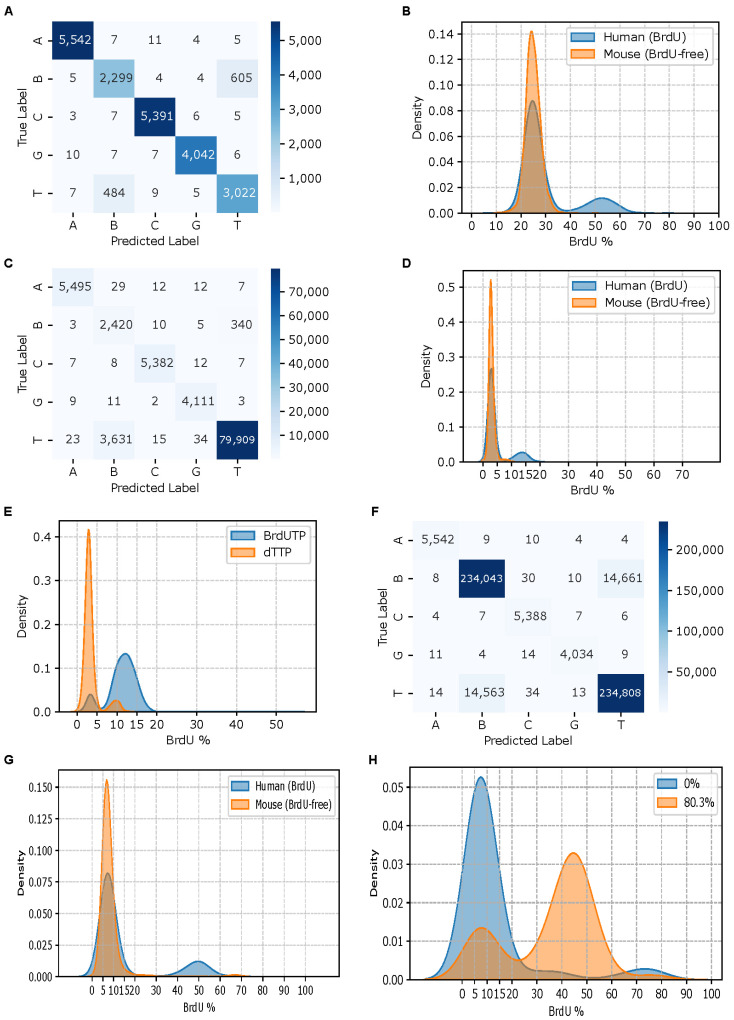
Performance of BiGRU-based RNN models for BrdU detection. (**A**) Heatmap showing the confusion matrix of Model I tested on held-out test data from the synthetic 5-mers reference standards. (**B**) BrdU substitution rate distributions predicted by Model I for individual reads from human HEK293 cells with a ~20% BrdU substitution rate and mouse 3T3 cells cultured without BrdU (Dataset III). (**C**) Heatmap showing the confusion matrix for Model II tested on held-out synthetic 5-mer standards (Dataset I) and BrdU-free mouse data (Dataset III). (**D**) BrdU substitution rate distributions predicted by Model II for reads from human HEK293 cells with a ~20% BrdU substitution rate and BrdU-free mouse 3T3 cells (Dataset III). (**E**) BrdU substitution rate distributions predicted by Model II for reads from DNA synthesized by primer extension without BrdUTP (0% BrdU) or with BrdUTP in place of dTTP (theoretical 100% BrdU). The BrdUTP sample was not entirely substituted, as a small subset of 0% BrdU reads likely originated from renatured plasmid DNA rather than primer extension (Dataset II). (**F**) Heatmap showing the confusion matrix for Model III tested on Datasets I and II, and BrdU-free mouse data (Dataset III). (**G**) BrdU substitution rate distributions predicted by Model III for reads from human HEK293 cells with a ~20% BrdU substitution rate and BrdU-free mouse 3T3 cells. (**H**) BrdU substitution rate distributions predicted by Model III for reads from MCM869 yeast culture with 0% and 80.3% BrdU substitution rates.

## Data Availability

The code used to preprocess Datasets I–IV, Jupiter notebooks for model training, and the three trained models are available at https://github.com/haibol2016/DeepBrdU (accessed on 1 November 2025). Training and testing data are available at Zenodo (https://doi.org/10.5281/zenodo.15593739).

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
