# Peer review of "Training Recurrent Neural Networks for BrdU Detection with Oxford Nanopore Sequencing: Guidance and Lessons Learned"

_genes, 2025, doi:10.3390/genes16111356_

Round 1

Reviewer 1 Report

Comments and Suggestions for Authors

The study addresses an interesting and relevant topic, namely the detection of BrdU in Nanopore sequencing data using deep learning methods. However, the manuscript lacks important methodological details and contains several weaknesses.

In the section Materials and Methods, the authors describe the datasets which were then used for model training and evaluation. However, here some important parameters of the sequencing data such as average read length, number of reads, quality of sequencing reads etc. are missing. Thus, without these significant parameters, it is not possible to assess the reliability and robustness of the datasets. These details should be provided for each dataset.

In section 2.5, the authors mention that the reads were mapped to several reference sequences, but the accession numbers or identifiers for these reference sequences are not provided.

In section 2.6, the authors state that the data were split into training, validation, and test sets, but the proportions of this division are not indicated. This is an important methodological detail, as the size of each subset has a direct impact on model performance and generalizability.

In the Results section, the authors acknowledge that their model does not outperform existing tools, mainly due to the limited amount of training data. In the Discussion, they further suggest strategies to expand the dataset. Why was this possibility of expanding the dataset not utilized? This point should be addressed.

The study was performed using the R9 nanopore chemistry, which is now outdated and has been replaced by the R10 chemistry. The authors should comment on how their approach would translate to R10.

The supplementary materials are incomplete. Figure S1 is referenced in the main text but is not included in the supplementary files.

Comments on the Quality of English Language

The overall quality should be improved. The manuscript contains a large number of typos and spelling mistakes. The text also contains awkward phrases that make it difficult to follow. I highly recommend that professional proofreading be carried out.

Here are some examples:
l. 23 – Google instead of Goolge
l. 131 –nucleotides instead of nucelotides
l. 145 – remaining instead of remainnig
l. 165 - originally instead of orignially
l. 173 - designated instead of desginated
l. 205 – generated instead of genereated
l. 213 - lengths instead of lengthes
l. 218 – datasets instead of datasetes
l. 237 – basecalled instead of basedcalled
l. 243 – derived instead of dervied
l. 246 – nucleotide instead of nuceotide
l. 256 – basecalled instead of basedcalled
l. 263 – corresponding instead of corresponing
l. 265 – random renaturation instead of radom renaturatioation

Author Response

The study addresses an interesting and relevant topic, namely the detection of BrdU in Nanopore sequencing data using deep learning methods. However, the manuscript lacks important methodological details and contains several weaknesses.

In the section Materials and Methods, the authors describe the datasets which were then used for model training and evaluation. However, here some important parameters of the sequencing data such as average read length, number of reads, quality of sequencing reads etc. are missing. Thus, without these significant parameters, it is not possible to assess the reliability and robustness of the datasets. These details should be provided for each dataset.

Response: We agree with the reviewer that data quality control is essential and that reliable research depends on the use of high-quality sequencing data. In the revised manuscript, we have clarified the quality control (QC) procedures and summarized key sequencing quality parameters for each dataset.

Quality control of the FASTQ data was performed for all datasets. For Datasets I, III, and IV, QC was conducted using the Nanopore sequencing–specific tool ToulligQC (v2.7.1; https://github.com/GenomiqueENS/toulligQC), while QC for Dataset II was performed using FastQC (v0.11.9; https://www.bioinformatics.babraham.ac.uk/projects/fastqc/). We have added the following statements to the revised manuscript:

  • Lines 210-211: “Quality control of reads in the FASTQ format was performed using ToulligQC (v2.7.1). The ToulligQC reports are available as Supplementary Materials (File S2).”
  • Lines 250-252: “For quality control of Dataset II, FASTQ reads corresponding to alignments with a SAM flag of 0 were extracted, and their quality was assessed using FastQC (v0.11.9). The resulting FastQC reports are provided in Supplementary Materials (File S2).”

A summary of QC results for each dataset has been added to the Supplementary Materials. In brief, Datasets I and III, as well as BO_testing_1 (BrdU-free, Dataset IV), exhibited overall good sequencing quality. Dataset II showed slightly lower read quality, averaging around Q7, likely because the basecalling was performed using an older basecaller (Metrichor). BY_test_1 (80.3% BrdU substitution rate, Dataset IV) contained approximately 72% of reads with quality scores below Q9.

Importantly, even though some raw data exhibited modest overall quality, we applied stringent filtering prior to model training (see Basecalling and Filtering and Resquiggling sections, Lines 202-252). After basecalling, all reads were aligned to the corresponding reference sequences, and reads with low mapping quality or shorter than the specified threshold were removed. For Dataset I, we additionally excluded reads lacking the expected five barcodes or 5-mer sequences. Reads that failed resquiggling were also excluded.

As a result, the final datasets used for model training, evaluation, and testing were of high quality and suitable for reliable model development and performance assessment.

Dataset

Sample description

Median read length (bp)

Number of reads

Median read sequencing quality PHRED score

Number of reads used for modeling

I

Synthetic 5-mer reference standard

372

1,037,284

12.6

215,226

II

BrdU-free

7,500

14,677

7

8,229

100% BrdU

7,500

5,868

6.8

3,142

III

Mouse (BrdU-free)

6,115

4,012

14.2

3,113

Human (20% BrdU substitution rate)

4,502

4,011

13.8

3,486

IV

BO_testing_1(BrdU-free)

12,541

4,001

15.8

3,201

BY_testing_1(80.3% BrdU substitute rate)

4,130

4,001

7.4

891

In section 2.5, the authors mention that the reads were mapped to several reference sequences, but the accession numbers or identifiers for these reference sequences are not provided.

Response: Thank you for pointing this out. We have now added detailed information about the reference sequences in Lines 213-218, as follows:
“The theoretical 5-mer synthesis sequences for Dataset I (Supplementary Materials, File S3); the reformatted pTYB21 plasmid sequence (
https://www.neb.com/en-us/tools-and-resources/interactive-tools/dna-sequences-and-maps-tool) for Dataset II; the Ensembl primary assembly of GRCm39 (mouse, GCA_000001635.9) and the Ensembl T2T-CHM13v2.0 (human, GCA_009914755.4) for Dataset III; and the Ensembl yeast reference genome R64-1-1 (GCA_000146045.2) for Dataset IV.”

In section 2.6, the authors state that the data were split into training, validation, and test sets, but the proportions of this division are not indicated. This is an important methodological detail, as the size of each subset has a direct impact on model performance and generalizability.

Response: These details were originally included in the Jupyter notebooks. We have now added the data-splitting information in the manuscript at Line 290: “…and split into training (80%), validation (10%), and test (10%) sets used for training.”

In the Results section, the authors acknowledge that their model does not outperform existing tools, mainly due to the limited amount of training data. In the Discussion, they further suggest strategies to expand the dataset. Why was this possibility of expanding the dataset not utilized? This point should be addressed.

Response: We appreciate the reviewer’s thoughtful comment. The main purpose of this manuscript is to provide guidance on data preparation and model architecture, and to share insights gained from our experience in building a basecaller. All available training datasets were generated using R9 flow cells. Generating additional data with the same (R9) chemistry would not be particularly meaningful, as these legacy flow cells have been discontinued. In the future, it would be ideal to generate BrdU-positive and BrdU-negative datasets using the latest Nanopore R10.4.1 flow cells. However, this effort is beyond the scope of the present study. Importantly, our data preparation pipeline and model architecture are designed to be transferable to R10 data or to other basecallers with only minimal modifications.

The study was performed using the R9 nanopore chemistry, which is now outdated and has been replaced by the R10 chemistry. The authors should comment on how their approach would translate to R10.

Response: Thank you for this thoughtful comment. We have clarified throughout the revised manuscript that our models were trained using data generated on legacy R9 flow cells and therefore cannot be directly applied to data generated on R10 flow cells. In the Discussion section, we explicitly emphasize this limitation and outline strategies to extend our approach to R10 (Lines 400-411):

“We acknowledge that our DeepBrdU models for BrdU detection were trained using Nanopore sequencing data generated on R9 flow cells with legacy chemistries, which have now been discontinued and replaced by R10 flow cells with updated chemistry. Given that the R10 flow cell features an improved design with two ‘readers’ along the nanopore and produces substantially different current signal patterns, our current model cannot be used to analyze data generated on R10 flow cells. We would have trained our model with BrdU-positive and BrdU-negative data generated on R10 flow cells if such training datasets were available. However, we believe that our methods for data preparation and model architecture can be readily extended to develop a new BrdU basecaller using R10-derived training data with only minor modifications: (1) using f5c (v1.2+) [35] or Uncalled4 [36] instead of Tombo for resquiggling, and (2) employing 9-mer rather than 5-mer features for model training [37].”

The supplementary materials are incomplete. Figure S1 is referenced in the main text but is not included in the supplementary files.

Response: Thank you for pointing out this issue. Figure S1 was originally submitted together with Figures 1 and 2 as PDF files; however, for some reason, it was not accessible to the reviewers. We have now included Figure S1, along with Files S1–S3, as part of the Supplementary Materials.

Comments on the Quality of English Language

The overall quality should be improved. The manuscript contains a large number of typos and spelling mistakes. The text also contains awkward phrases that make it difficult to follow. I highly recommend that professional proofreading be carried out.

Here are some examples:
l. 23 – Google instead of Goolge
l. 131 –nucleotides instead of nucelotides
l. 145 – remaining instead of remainnig
l. 165 - originally instead of orignially
l. 173 - designated instead of desginated
l. 205 – generated instead of genereated
l. 213 - lengths instead of lengthes
l. 218 – datasets instead of datasetes
l. 237 – basecalled instead of basedcalled
l. 243 – derived instead of dervied
l. 246 – nucleotide instead of nuceotide
l. 256 – basecalled instead of basedcalled
l. 263 – corresponding instead of corresponing
l. 265 – random renaturation instead of radom renaturatioation

Response: Thank you very much for your comments on the writing. We have now corrected all spelling errors, made multiple minor edits throughout the manuscript, and carefully proofread the entire text.

Reviewer 2 Report

Comments and Suggestions for Authors

This manuscript presents a detailed protocol for preparing training data and implementing a recurrent neural network to detect BrdU incorporation sites from nanopore sequencing data. The authors address a recognized gap in the field: the lack of accessible tutorials for applying deep learning to this specific biological problem. The work is highly practical and aims to facilitate broader adoption of these techniques by the research community. Overall, this work is well designed, and the tool will benefit the community. I have two major concerns for consideration.

The first one is the applicability to current sequencing chemistry. A significant limitation is that the training and testing datasets are based on the R9 flow cell, which is now legacy chemistry and no longer supported by Oxford Nanopore Technologies (ONT). The current chemistry, the R10 flow cell, has a different nanopore design and produces distinctly different signal signatures. Therefore, the model as presented is not readily applicable to data generated with modern chemistry. The authors should explicitly state in the Abstract that the results are based on R9 chemistry. Furthermore, the Discussion section must be expanded to acknowledge this limitation. The authors should discuss the implications and outline how the general strategy of their protocol (data preparation, model architecture) could be adapted for R10 data. If any publicly available BrdU sequencing data generated with R10 chemistry exists, an effort should be made to include it for validation or to discuss its potential use in future work.

The second one is clarification in performance evaluation. The description of the model's performance metrics (specificity >94%, moderate sensitivity) lacks critical context. It is unclear at what level of resolution and on what unit of data these metrics are calculated. Is the model making predictions at single-nucleotide (base) resolution, or for k-mers/motifs? Are the sensitivity and specificity values calculated per individual sequencing read (single-molecule level) or are they aggregated across all reads from a sample (bulk-molecule level)? This distinction is crucial for interpreting the tool's practical utility.

Author Response

This manuscript presents a detailed protocol for preparing training data and implementing a recurrent neural network to detect BrdU incorporation sites from nanopore sequencing data. The authors address a recognized gap in the field: the lack of accessible tutorials for applying deep learning to this specific biological problem. The work is highly practical and aims to facilitate broader adoption of these techniques by the research community. Overall, this work is well designed, and the tool will benefit the community. I have two major concerns for consideration.

Response: Thank you very much for your positive comments.

The first one is the applicability to current sequencing chemistry. A significant limitation is that the training and testing datasets are based on the R9 flow cell, which is now legacy chemistry and no longer supported by Oxford Nanopore Technologies (ONT). The current chemistry, the R10 flow cell, has a different nanopore design and produces distinctly different signal signatures. Therefore, the model as presented is not readily applicable to data generated with modern chemistry. The authors should explicitly state in the Abstract that the results are based on R9 chemistry. Furthermore, the Discussion section must be expanded to acknowledge this limitation. The authors should discuss the implications and outline how the general strategy of their protocol (data preparation, model architecture) could be adapted for R10 data. If any publicly available BrdU sequencing data generated with R10 chemistry exists, an effort should be made to include it for validation or to discuss its potential use in future work.

Response: Thank you for your thoughtful and constructive comments. We fully recognize that Oxford Nanopore Technologies (ONT) has discontinued the R9 flow cells, and that our models, trained using data generated on R9.4 flow cells, are not directly applicable to data produced using the newer R10 chemistry. However, we believe that our data preparation pipeline and model architecture can be readily adapted to train new models once BrdU-positive and BrdU-negative datasets generated on R10 flow cells become available.

In response to the reviewer’s suggestion, we have explicitly clarified this limitation and its implications throughout the revised manuscript:

  • Abstract (Lines 18–21): “Due to the lack of ground truth BrdU-positive data generated on the latest Nanopore R10 flow cells, we prepared model training data from legacy Nanopore R9 flow cells, consistent with existing tools.”
  • Abstract (Line 28): “Though the models were trained with data generated on legacy flow cells, …”
  • Introduction (Lines 119-121): “Due to unavailability of ground truth BrdU-positive data generated on the latest Nanopore R10 flow cells, we prepared model training data from legacy Nanopore R9 flow cells.”
  • Discussion (Lines 400-411): Expanded discussion emphasizing this limitation and describing potential adaptations for R10, including (1) using f5c (v1.2+) or Uncalled4 for resquiggling, and (2) employing 9-mer features for model training.
  • Conclusion (Lines 437-439): Added statements acknowledging this limitation and reiterating the future applicability of our approach to R10 chemistry.

Unfortunately, no publicly available BrdU sequencing data generated using R10 chemistry are currently available. We have noted this in the Discussion and highlighted it as a key direction for future work.

The second one is clarification in performance evaluation. The description of the model's performance metrics (specificity >94%, moderate sensitivity) lacks critical context. It is unclear at what level of resolution and on what unit of data these metrics are calculated. Is the model making predictions at single-nucleotide (base) resolution, or for k-mers/motifs? Are the sensitivity and specificity values calculated per individual sequencing read (single-molecule level) or are they aggregated across all reads from a sample (bulk-molecule level)? This distinction is crucial for interpreting the tool's practical utility.

Response: Thank you for pointing out this important issue. We have now added a new subsection, Section 2.7: Performance Metrics, in the Materials and Methods (Lines 303-313) to clarify the level of resolution and units used for performance evaluation. The added text reads as follows:

“Performance metrics of the models were calculated using the following equations: accuracy = (TP + TN)/(TP + TN + FP + FN), precision = TP/(TP + FP), sensitivity (recall) = TP/(TP + FN), and specificity = TN/(TN + FP), where TP, TN, FP, and FN represent true positives, true negatives, false positives, and false negatives, respectively. All performance metrics were evaluated at single-nucleotide resolution, using features for individual 5-mers as input and predicted base types at the central positions of the 5-mers as output. The BrdU substitution rate was calculated at the read level as the percentage of thymidine bases in a read predicted as BrdU by the models.”

Round 2

Reviewer 1 Report

Comments and Suggestions for Authors

I suggest writing "Sequencing" and "Nanopore" in lowercase (l. 16, 18, 19 etc.) when it is not part of the company name.

The term "fast5" should be written in lowercase when used as a file suffix (l. 175, 176).

The figure on page 5 appears to be identical to the one on page 6.

In the headings of sections 3.1, 3.2, and 3.3, I recommend not capitalizing every word.

Regarding the limited dataset, would it be possible to use nanopore data simulators to augment the training data? This could help improve model sensitivity and generalizability.

Author Response

We appreciate the reviewer’s thoughtful, detailed comments, which have significantly enhanced the manuscript’s clarity and readability.

I suggest writing "Sequencing" and "Nanopore" in lowercase (l. 16, 18, 19 etc.) when it is not part of the company name.

Response: All occurrences of “Sequencing” and “Nanopore” except the ones as part of the company name are in lowercase now.

The term "fast5" should be written in lowercase when used as a file suffix (l. 175, 176).

Response: the two occurrences of “FAST5” have been changed to “fast5”.

The figure on page 5 appears to be identical to the one on page 6.

Response: Now only one is included in the manuscript.

In the headings of sections 3.1, 3.2, and 3.3, I recommend not capitalizing every word.

Response: Now, only the first word is capitalized in the section headings 3.1, 3.2, and 3.3.

Regarding the limited dataset, would it be possible to use nanopore data simulators to augment the training data? This could help improve model sensitivity and generalizability.

Response: We thank the reviewer for this thoughtful suggestion. Indeed, several simulation tools can be used for generating Oxford nanopore DNA sequencing data with corresponding raw current signals, including Squigulator (https://genome.cshlp.org/content/34/5/778), seq2squiggle (https://doi.org/10.1093/bioinformatics/btae744), Icarust (https://doi.org/10.1093/bioinformatics/btae141), and DeepSimulator (doi: 10.1093/bioinformatics/bty223). However, existing simulators are calibrated to current signals from standard nanopore DNA sequencing and do not model the BrdU-induced signal shifts. As a result, their simulated reads would not faithfully capture the modified current profiles needed for our task. We therefore did not use nanopore simulators for data augmentation and instead trained and validated solely on experimentally derived, ground truth BrdU-positive reads, which provides a more biologically representative basis.